# High-resolution cryo-EM structure of urease from the pathogen *Yersinia enterocolitica*

Ricardo D. Righetto [1,4], Leonie Anton [2,4], Ricardo Adaixo[1,4], Roman P. Jakob [2], Jasenko Zivanov[1], Mohamed-Ali Mahi[2,3], Philippe Ringler[1], Torsten Schwede [2,3], Timm Maier[2✉] & Henning Stahlberg[1✉]

Urease converts urea into ammonia and carbon dioxide and makes urea available as a nitrogen source for all forms of life except animals. In human bacterial pathogens, ureases also aid in the invasion of acidic environments such as the stomach by raising the surrounding pH. Here, we report the structure of urease from the pathogen *Yersinia enterocolitica* at 2 Å resolution from cryo-electron microscopy. *Y. enterocolitica* urease is a dodecameric assembly of a trimer of three protein chains, ureA, ureB and ureC. The high data quality enables detailed visualization of the urease bimetal active site and of the impact of radiation damage. The obtained structure is of sufficient quality to support drug development efforts.

[1] Center for Cellular Imaging and NanoAnalytics, Biozentrum, University of Basel, Mattenstrasse 26, CH-4058 Basel, Switzerland. [2] Biozentrum, University of Basel, Klingelbergstrasse 50/70, CH-4056 Basel, Switzerland. [3] SIB Swiss Institute of Bioinformatics, Biozentrum, University of Basel, Klingelbergstrasse 50/70, CH-4056 Basel, Switzerland. [4]These authors contributed equally: Ricardo D. Righetto, Leonie Anton, Ricardo Adaixo. ✉email: timm.maier@unibas.ch; henning.stahlberg@unibas.ch

Ureases are nickel-metalloenzymes produced in plants, fungi, and bacteria, but not in animals. They facilitate nitrogen fixation by metabolizing urea, but in some pathogenic bacteria they serve a dual function and consume ammonia to promote survival in acidic environments. This function is vital during host infection where the bacteria have to survive the low pH of the stomach[1], and of phagosomes in host cells[2–4]. The relevance of ureases in early stages of infection renders them attractive targets for novel anti-microbials.

Ureases catalyze the breakdown of urea into ammonia and carbamate at a rate $10^{14}–10^{15}$ times faster than the non-catalyzed reaction (Fig. 1a), and are arguably the most efficient hydrolases[3,4]. In a second non-catalyzed step, carbamate is spontaneously hydrolyzed to yield another molecule of ammonia as well as one molecule of bicarbonate[4]. Jack bean urease was the first enzyme to be crystallized, offering evidence that enzymes are proteins[5] and was the first metalloenzyme to be shown to use nickel in its active site[6]. All ureases characterized to date share the architecture of their active site[4,7]. Two $Ni^{2+}$ ions in the active site are coordinated by a carbamylated lysine, four histidines, and one aspartate. The two metal ions are bridged by a hydroxide ion, serving as a nucleophile[8,9]. Urea first interacts with Ni(1) through its carbonyl oxygen, and following a conformational change of a mobile flap covering the active site, one of the amino nitrogens then binds to Ni(2), making the resulting metal-chelate urea molecule more available for nucleophilic attacks. Subsequently, the bridging hydroxide acts as the nucleophile attacking the urea C atom, and a proton is then transferred from the hydroxide ion to the distal $NH_2$ group, yielding a nascent ammonia molecule. The latter is released upon breaking of the $C–NH_3^+$ bond, yielding carbamate as the byproduct, the latter spontaneously undergoing hydrolysis. The active site is closed off during the reaction by a conformationally variable helix-turn-helix motif, referred to as the mobile flap. Despite their common catalytic mechanism, ureases display different chain topologies and higher order oligomeric assemblies (Fig. 1b, c). Jack bean urease assembles into an oligomer of a single type of polypeptide chain with D3 symmetry ($[[\alpha]_3]_2$ D3) (Fig. 1b, c)[3,4]. The urease of the human pathogen *Helicobacter pylori* is composed of two types of polypeptide chains (ureA, ureB) and assembles into a dodecamer

(tetramer-of-trimers) with tetrahedral symmetry ($[[\alpha\beta]_3]_4$ T) (Fig. 1b, c). The ability of ureases to raise the pH of their environment benefits *H. pylori* in colonizing the stomach and downstream gut, causing gastric ulcers[7]. The urease of the opportunistic pathogen *Klebsiella aerogenes*[10] assembles into a heterotrimer of three proteins ureA, ureB, and ureC, which in turn oligomerizes into a trimer ($[\alpha\beta\gamma]_3$ C3)[11] (Fig. 1b, c). This oligomeric assembly is common also to most other structurally characterized bacterial ureases[3,4].

*Yersinia enterocolitica* is the causative agent of yersiniosis, a gastrointestinal infection, reactive arthritis, and erythema nodosum. The infection spreads to humans through consumption of contaminated food with pigs being one of the largest reservoirs of *Y. enterocolitica*. The symptoms of yersiniosis are fever, abdominal pain, diarrhea, and/or vomiting and is one of the most reported enteritis in some countries, although outbreaks are rare[12]. *Y. enterocolitica* is a facultative intracellular bacterium and can survive in very different environments. In the presence of urea *Y. enterocolitica* can tolerate extremely acidic conditions of pH 1.5[2]. *Y. enterocolitica* urease comprises three polypeptide chains (ureA, ureB, and ureC), an architecture similar to that of *K. aerogenes* urease (Fig. 1b) and has been biochemically characterized previously[13].

Here we present the structure of *Y. enterocolitica* urease at an overall resolution of 2 Å, which was achieved using recent advances in cryo-EM data collection and processing. The structure shows that *Y. enterocolitica* urease assembles into a dodecameric hollow sphere with a $[[\alpha\beta\gamma]_3]_4$ oligomeric assembly structure of tetrahedral symmetry. A tightly embedded kinked loop is interacting with neighboring domains and is potentially responsible for the assembly of the oligomer. The data allows model building of the active site carbamylated lysine, and visualization of radiation damage to the nickel-metal center as well as of hydration networks throughout the protein.

## Results

**Structure determination of *Y. enterocolitica* urease by cryo-EM.** We have used single particle cryo-EM to determine the structure of the fully assembled *Y. enterocolitica* urease. We acquired 4494

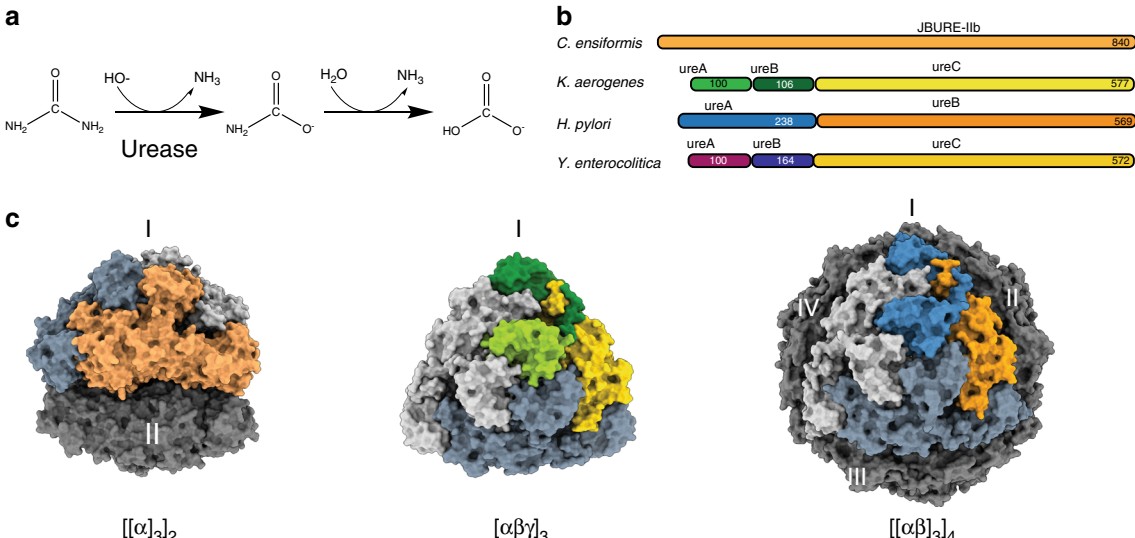

**Fig. 1 Protein architectures and oligomeric assemblies of ureases. a** Schematic of biochemical reaction catalyzed by urease. **b** Protein architecture of urease functional unit from full length *C. ensiformis* (Jack bean urease)[58], *K. aerogenes*, *H. pylori*, and *Y. enterocolitica*. **c** Surface representation of oligomeric assembly of urease in *C. ensiformis* (PDB: 3LA4), *K. aerogenes* (PDB: 1EJW), *H. pylori* (PDB: 1E9Z), respectively. Proteins are color-coded as in **b**. Oligomeric state of urease assembly is indicated at the bottom and the trimeric assemblies are indicated in roman numerals.

**Table 1 Cryo-EM data collection, refinement and validation statistics.**

|  | Urease (EMDB-10835) (PDB 6YL3) |
|---|---|
| Data collection and processing |  |
| Magnification | 78,247 (215,000 nominal) |
| Voltage (kV) | 300 |
| Electron exposure (e−/Å²) | 42 |
| Defocus range (μm) | −0.2 to −1.5 |
| Pixel size (Å) | 0.639 |
| Symmetry imposed | T |
| Initial particle images (no.) | 194,603 |
| Final particle images (no.) | 119,020 |
| Map resolution (Å) | 1.98 |
| FSC threshold | 0.143 |
| Map resolution range (Å) | 1.94–2.40 |
| Refinement |  |
| Initial model used (PDB code) | 4Z42 |
| Model resolution (Å) | 2.02 |
| FSC threshold | 0.5 |
| Model resolution range (Å) | — |
| Map sharpening $B$ factor (Å²) | −38 |
| Model composition |  |
| Non-hydrogen atoms | 77,076 |
| Protein residues | 9,552 |
| Ligands | 24 |
| $B$ factors (Å²) |  |
| Protein | 14.5 |
| Ligand | 22.9 |
| R.M.S. deviations |  |
| Bond lengths (Å) | 0.066 |
| Bond angles (°) | 3.665 |
| Validation |  |
| MolProbity score | 1.93 |
| Clashscore | 4.37 |
| Poor rotamers (%) | 2.74 |
| Ramachandran plot |  |
| Favored (%) | 94.35 |
| Allowed (%) | 5.27 |
| Disallowed (%) | 0.38 |

movies of urease particles using a Titan Krios transmission electron microscope (TEM) equipped with a K2 direct electron detector and an energy filter (see "Methods" for details). Approximately half of the movies (2243) were acquired by illuminating three locations (shots) per grid hole using beam-image shift in order to speed up the data collection[14], whereas the remaining movies were recorded without this feature i.e., just a single shot at the center of the hole. This allowed us to measure and assess the extent of beam tilt and other optical aberrations, as well as the behavior of sample drift between each condition and beam-image shift position. Typical micrographs from the imaged grids are shown in Supplementary Fig. S1a and a summary of data collection information is given in Table 1.

Each dataset was processed separately for 3D reconstruction following the strategy depicted in Supplementary Fig. S2. The first obtained 3D map, at an overall resolution of 2.6 Å, revealed that this urease assembly is a dodecamer of tetrahedral (T) symmetry with a diameter of ~170 Å. The separate processing of each dataset yielded refined 3D maps at nominal resolutions of 2.10 Å and 2.20 Å for the multi-shot and single−shot cases, respectively (see "Methods").

For comparison, we also processed the merged set of particles from both datasets altogether. We observed on the 2D class averages a preferential orientation for the threefold symmetric view of urease, and also the presence of isolated monomers and broken assemblies (Supplementary Fig. S1b). The presence of

such incomplete assemblies was further confirmed by performing 3D classification without imposing symmetry, as shown in Supplementary Fig. S1c. The 3D class corresponding to the complete dodecameric assembly of urease contained 119,020 particles, of which 69,512 (58.4%) came from the multi-shot and 49,518 (41.6%) from the single-shot dataset. With respect to the number of particles picked from each dataset, 64.7% of the particles from the multi-shot and 56.8% from the single-shot datasets were retained at this stage and throughout the final reconstruction. While coma-free alignment was performed and active beam-tilt compensation in SerialEM[15] was used on our data collections, after performing beam tilt refinement in RELION-3[16] we observed that the single-shot case has a residual beam-tilt higher than the smallest residual observed in the multi-shot case (Supplementary Table S2). These two values are however very close to zero and are possibly within the error margin of the post hoc beam tilt refinement procedure.

The reduced need to move the specimen stage in beam-image shift mode not only speeds up data collection but also minimizes stage drift. The second and third shots from the multi-shot dataset have comparatively less drift than both the first multi-shot and the single shot, as suggested by the parameter values obtained from the Bayesian polishing training[17] on each beam-tilt class separately (Supplementary Table S3). As all the three multi-shots are taken in nearby areas within the same foil hole, this observation is consistent with movement by the annealing behavior of the vitreous ice layer and its carbon support after pre-irradiating the specimen as reported previously[18].

At this point, the nominal resolution of the map after 3D refinement was 2.05 Å. Finally, correcting for residual higher-order aberrations in CTF refinement[19] (Supplementary Fig. S3) yielded a map at a global resolution of 1.98 Å (Supplementary Fig. S4a). Local resolution estimation reveals that the core of the map is indeed at this resolution level or better (Fig. 2a and Supplementary Fig. S4b), and the local resolution-filtered map was then used for model building as explained in the next section. Despite the 12-fold symmetry of the urease assembly, a limiting factor in the resolution of the map is the strong presence of preferential orientation, as confirmed by the plot of the final orientation assignments (Supplementary Fig. S4c). The estimated angular distribution efficiency is 0.78[20]. An overview of the cryo-EM map and its main features are depicted in Supplementary Movie 1.

**Y. enterocolitica urease assembles as a tetramer of trimers.** Model building was initiated from available crystallographic models with subsequent fitting and refinement against the cryo-EM map. The model was built and refined for one asymmetric unit containing one copy of the ureA, ureB, and ureC protein each. The model was then expanded using NCS (see "Methods"). The complete model covering the whole oligomeric assembly contains 9552 residues, 3672 waters and 24 nickel ions (two per active site, twelve active sites) (Table 1). The quality of the model was assessed with the cryo-EM validation tools in the PHENIX package[21]. The map allowed for the building of all residues of ureA (1–100), and residues 31–162 of ureB and 2-327/335–572 of ureC (Supplementary Fig. S5). The hetero-trimer formed by the three protein chains (ureA, ureB, ureC) (Fig. 1b) oligomerizes into a homo-trimer. The homo-trimer is arranged in a tetramer-of-trimers making the full complex a dodecamer of the hetero-trimer (Fig. 2b). There are four large oval shaped holes between the trimers (64 Å long, 12 Å wide, high electrostatic potential) and four smaller holes at the center of the trimer with a diameter of 6 Å (low electrostatic potential), as shown in Fig. 2b, and the center of the enzyme assembly is hollow (Fig. 2). The holes

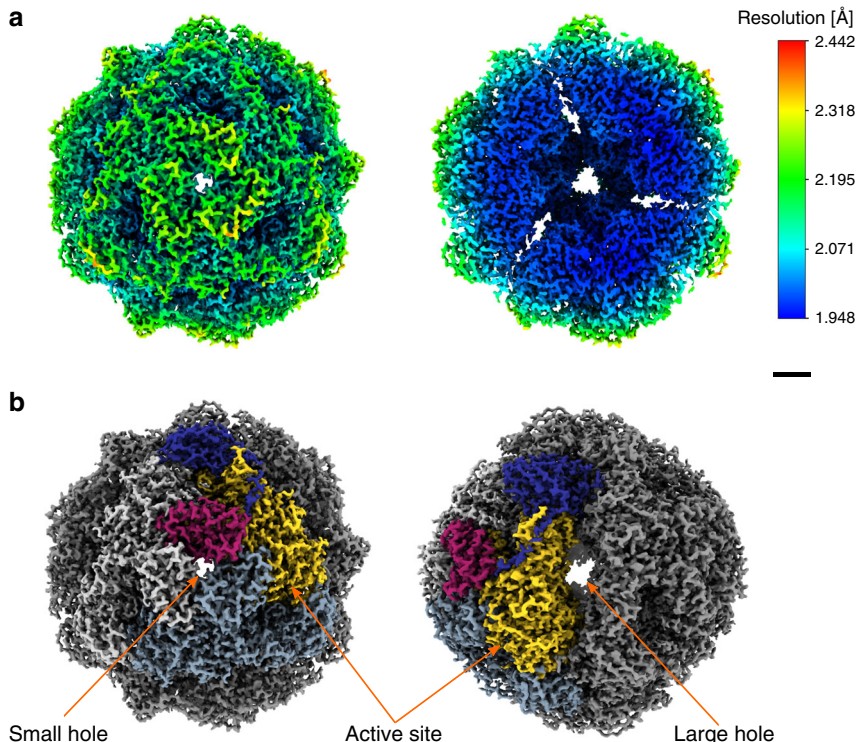

**Fig. 2 Cryo-EM analysis of the *Y. enterocolitica* dodecameric urease assembly. a** The cryo-EM map (left) filtered and colored by local resolution and a slice cut through the map (right) to show the internal details. **b** The assembly architecture highlighted on the map. The three chains that form the basic hetero-trimer are shown in different colors, with the other hetero-trimers shown in shades of gray. Two different views are shown to indicate the location of the small and larger holes at the interfaces, as well as the active site. Scale bars: 20 Å.

provide ample opportunity for diffusion of the uncharged substrate and product, and the hollow inside potentially leads to a local increase of reaction product. The assembly has the same symmetry as the urease homolog in *H. pylori*, which was postulated to increase stability and/or resistance to acidic environments[22].

For analysis of the protein sequences, the ConSurf Server[23,24] was used with the sample list of homologs option to get a diverse set of 150 sequences. The protein chains of *Y. enterocolitica* urease are highly conserved across different organisms. The ureA chain is split after a LVTXXXP motif and is 99–100 amino acids long in most cases, with a sequence identity of 55.7%. The ureB chain of *Y. enterocolitica* has between 20 and 30 N-terminal amino acids more compared to the other sequences (except *Kaistia sp.* SCN 65-12), which share an identity of 51.5%. This N-terminal extension is located on the outside of the holoenzyme where ureA and ureB chain split occurs and are too disordered to be modeled in the structure (Supplementary Fig. S5). The charges and properties of this stretch of amino acids vary and if they still serve a function remains unclear. The last 20 amino acids of the C-terminus of ureB are only represented in half of the compared sequences and accurate sequence conservation could not be determined in this part. This stretch contains a loop and a C-terminal helix (Supplementary Fig. 5). The ureC protein of the compared sequences has a shared sequence identity of 60.3%. All amino acids involved in catalysis are highly conserved (Supplementary Fig. S5). The ureA and ureB chains show lower conservation compared to ureC. They are not involved in catalysis but in scaffolding, so the differences could stem from their role in different types of oligomeric assembly (Fig. 3).

To investigate this aspect further, we compared the presented structure to the ureases of *H. pylori*, *S. pasteurii*, and *K. aerogenes*. Sequence identity scores among these ureases are provided in Supplementary Table S4. The *H. pylori* urease is made up of two protein chains ureA (that contains the equivalent of ureA and ureB in *Y. enterocolitica*) and ureB (that is the equivalent of ureC in *Y. enterocolitica*) (Fig. 1c and Fig. 3c). It assembles into a T-symmetric oligomer like in *Y. enterocolitica* and the crystal structure was solved to 3 Å. *S. pasteurii* and *K. aerogenes* ureases both assemble into a trimer from the hetero-trimeric unit (Figs. 1c, 3b). *S. pasteurii* urease has been solved by X-ray crystallography in different conditions (for example, PDB IDs: 2UBP, 3UBP, 4CEU)[25,26]. Here we use the highest resolution urease structure, that was solved in the presence of the inhibitor N-(n-Butyl)thiophosphoric Triamid (NBPT) to 1.28 Å, for comparison (PDB ID: 5OL4)[27]. The crystal structure of *K. aerogenes* used for comparison in this paper has a similar resolution range and was solved to 1.9 Å in absence of substrate or inhibitors (PDB ID: 1EJW)[11].

There are two main regions with high root mean square deviations (RMSDs) when comparing these three ureases to the *Y. enterocolitica* model (Supplementary Fig. S6 and Supplementary Table S5). The first region with high deviation is the mobile flap, which opens and closes over the active site (residues 312–355 of ureC). Both the open and the closed conformations of the mobile flap have been observed in crystal structures, stabilized at pH values lower and higher than the pKa of the conserved His323, respectively[8]. The residues of its connecting loop could not be built with confidence in the cryo-EM model (residues 326–333 of ureC). The other region with large differences is on the edges of ureA and ureB where the interactions with the next protomer occur. The *H. pylori* assembly contains an additional C-terminal loop (residues 224–238 of ureA) after the top alpha helix (residues 206–223 of ureA). This helix (central helix) forms the threefold axis of three neighboring trimers and the loop binds in a head-to-tail fashion to the next trimer forming the tetramer

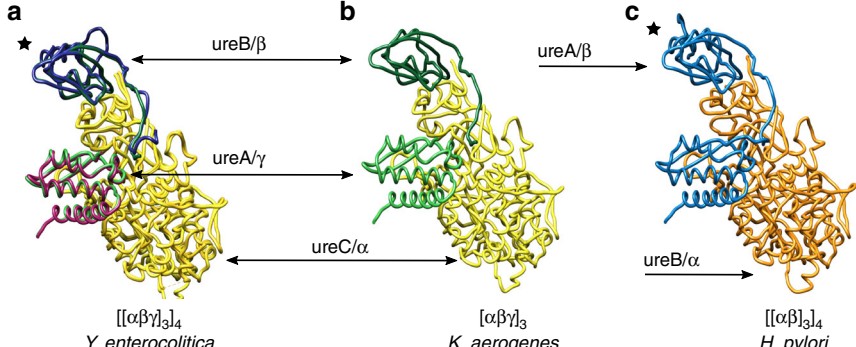

**Fig. 3 Comparison of *Y. enterocolitica* urease chain architecture with ureases with different modes of assembly from other pathogens.** Hetero-trimers are shown in tube representation with each chain in the same colors of the sequences in Fig. 1b. The black star indicates the central helix of the β subunit. The *Y. enterocolitica* and *H. pylori* ureases form the same dodecameric assembly despite having different types of chain splitting, while *K. aerogenes* urease has the same type of chain splitting as in *Y. enterocolitica* but forms only a trimeric assembly.

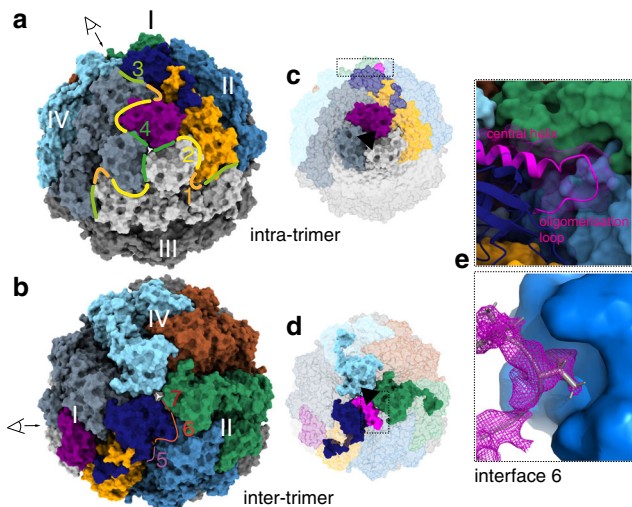

**Fig. 4 Interfaces in dodecameric assembly of *Y. enterocolitica*. a** surface model in front view of trimer of *Y. enterocolitica* urease with intra-trimeric interfaces 1–4 indicated with color-coded lines and numbers. **b** Same model shown from the top (view indicated with eye) and the inter-trimeric interfaces. **c** Front view with intra-trimeric-core highlighted and threefold axis indicated with black triangle. Inset for **e** in dashed box. **d** same as **c** but from the top view. **e** Interface 6 with loop from ureB (magenta) binding into pocket of ureC of neighboring trimer. Upper inset shows ureB in cartoon and transparent surface and ureC in surface representation. Lower panel shows ureC as surface and ureB loop as cartoon with density.

(Fig. 3a, c)[22]. The core of the assembly is identical in its structure. For whole-chain superposition scores and RMSD values between the compared models please see Supplementary Table S5.

In the dodecameric assembly seven different interfaces are formed between the hetero-trimers (Fig. 4a, b and Supplementary Fig. S7). Intra-trimer interactions occur between the three basic hetero-trimers in one assembled trimer, forming a threefold symmetry axis (Fig. 4a). The interactions between these trimers to form the tetramer then make up a different threefold symmetry axis (Fig. 4b). The three largest interfaces (interfaces 1–3) are formed intra-trimeric between ureC of one hetero-trimer and ureC, ureA, and ureB of the next trimer (Fig. 4a). The three ureA proteins make up the intra-trimer-core (first threefold axis) with interface 4 (Fig. 4c and Supplementary Fig. S7). Comparison of the interface areas formed in the trimer assembly shows no substantial differences between the four organisms (Supplementary Fig. S7). Inter-trimer interfaces (interfaces 5, 6, 7) formed in

the dodecameric *Y. enterocolitica* and *H. pylori* ureases have similar areas (Fig. 4b and Supplementary Fig. S7). Part of the interactions occur between ureB and ureC forming interfaces with each other (interface 4, 6). The other interaction is between the three ureB proteins and forms interface 7 and the inter-trimer-core (second threefold axis) with their central helices (Fig. 4b). *Y. enterocolitica* does not have the same oligomerization loop after the central helix proposed for *H. pylori*. However, there is a short loop before the central helix, which is extended in *Y. enterocolitica*. It binds into a pocket of ureC of the neighboring trimer in interface 6 (Fig. 4e). These types of loops or extensions are missing from *S. pasteurii* and *K. aerogenes* ureB proteins. *S. pasteurii* has the central helix, but there is no extended loop before or after it (Supplementary Fig. S6b). *K. aerogenes* urease does not have a helix nor a loop in this region (Fig. 3b). This suggests that the presence of oligomerization loops in ureB is crucial for determining the oligomeric state of the enzyme.

The dodecameric holoenzyme structure of ureases might aid in stabilizing the protein at acidic pH, and in combination with 12 active sites producing ammonia enables the formation of a pH-neutralizing microenvironment around the assembly[22]. This ensures the continued function of the enzyme and makes this type of oligomeric assembly essential to survival of *Y. enterocolitica* in the host. It is remarkable that *Y. enterocolitica* is the first organism outside the Helicobacteraceae family to have a known dodecameric urease. Considering the different subunit organization between these ureases, it raises the question of what particular events in the evolutionary history of *Y. enterocolitica* could have led to this type of assembly[28].

**The empty active site is filled with water.** At the global resolution of 1.98 Å, detailed structural features can be observed. All throughout the highly resolved areas of the protein, salt bridges, backbone and side chain hydration, and alternative side chain conformations can be visualized (Supplementary Fig. S8a–c). Furthermore, the high resolution allows for a detailed description of the nickel-metallo-center and the active site. The active site is located on the ureC protein at the edge of the hetero-trimer and is wedged in between the ureA and ureB proteins of the next hetero-trimer in the homo-trimeric assembly (Fig. 5a).

The catalysis of ammonia and carbamate from urea occurs in two steps (Fig. 1a). Urea first interacts with the nickel ions through its carbonyl oxygen and amino nitrogens. The active site contains two $Ni^{2+}$ ions which are coordinated by six different amino acids (Fig. 5b). Both $Ni^{2+}$ ions are coordinated by the carbamylated Lys222*. Ni(1) is additionally coordinated by His224, His251 and His277 and Ni(2) by His139, His141, and Asp365. Close to the

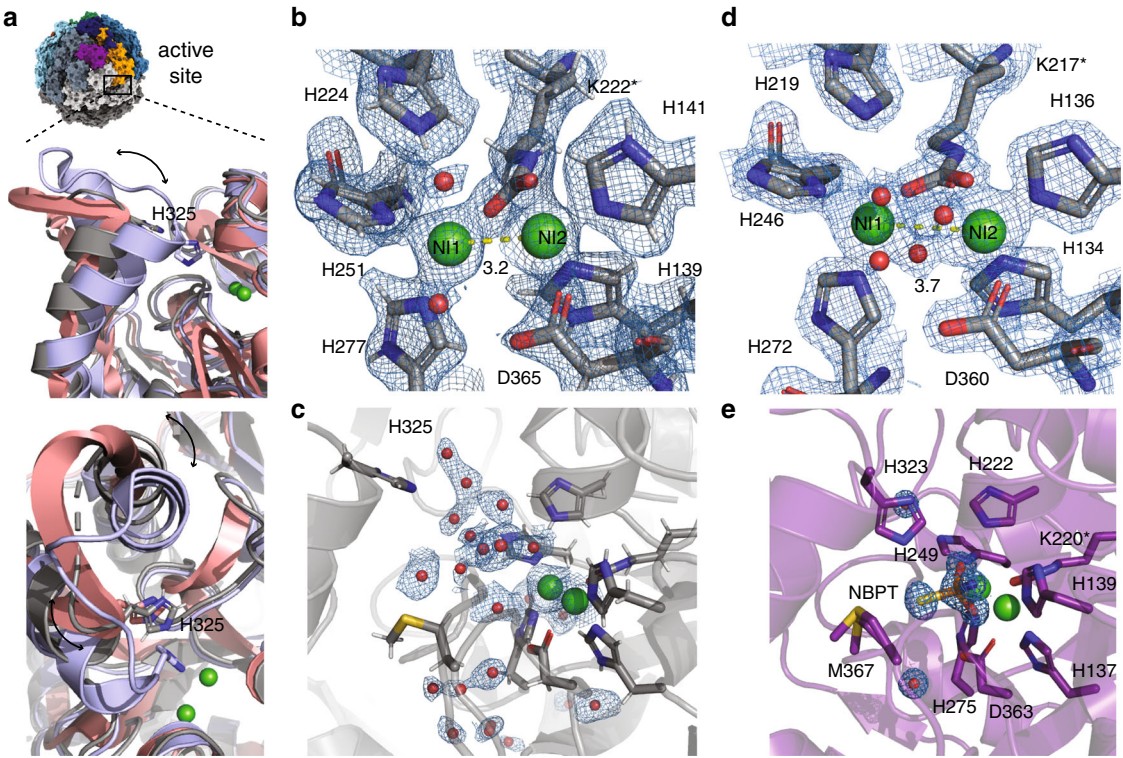

**Fig. 5 Active site of *Y. enterocolitica* urease. a** Overview of urease assembly with the active site location indicated. Inset shows in top panel side view of urease crystal structures from *S. pasteurii* mobile flap shown in open conformation in salmon (PDB: 2UBP) and in closed position as light purple (PDB: 3UBP). In gray the cryo-EM structure of *Y. enterocolitica* is overlaid and the green spheres represent the $Ni^{2+}$ ions of the active site. Bottom panel shows top view of the three structures. Arrows indicate movement of helix and catalytic His325 is shown as stick. **b** Model of active site residues and $Ni^{2+}$ ions with the cryo-EM map of *Y. enterocolitica* at 1.98 Å nominal resolution. Yellow line indicates distance between $Ni^{2+}$ ions in Å; **c** shows the water molecules in the active site. **d** Crystal structure of *K. aerogenes* urease at 1.9 Å resolution (PDB: 1EJW). Yellow line indicates distance between $Ni^{2+}$ ions in Å. **e** Crystal structure of *S. pasteurii* at 1.28 Å with inhibitor NBPT (PDB: 5OL4).

active site is a methionine (Met369), which can be modeled in different alternative conformations. One conformation could potentially reach the active site. There is no described function for this amino acid (Fig. 5c and Supplementary Fig. S9).

The active site is protected by a helix-turn-helix motif, called the mobile-flap. Its function is to coordinate the access of substrate to the catalytic site and the release of the product from it[3,4]. The protonation state of a conserved histidine on the mobile flap (His325) is essential for catalysis by determining opening and closing of the mobile flap and strongly depends on the solution pH[4,8] (Fig. 5a). By closing of the mobile flap His325 moves closer to the active site, stabilizing the distal amine of urea in the active site pocket[3,4,8,9]. After closing of the mobile flap, the urea molecule chelates the two Ni ions in the active site, and following the nucleophilic attack by the bridging hydroxide onto the urea C atom, a proton is transferred to the distal amine group from the metal-bridging C–OH group, yielding an ammonia molecule after breakage of the resulting C-NH3+ bond. Flap opening then releases ammonia and carbamate, where the latter spontaneously hydrolyzes into another molecule of ammonia and bicarbonate. The mobile flap of the cryo-EM structure presented here is modeled in an open position, however the local resolution is lower than in the surrounding areas, indicating flexibility. The sample was frozen in a buffer of pH 7.0, where the mobile flap of urease can adopt both open and closed conformations[8]. The twelve active sites of each particle adopting different conformations are averaged by single particle reconstruction with symmetry imposition into an mainly open conformation. In the absence of substrate or inhibitors in the sample, the mobile flap cannot be stabilized in one conformation (Fig. 5a). Coordinated

water molecules can be seen in the empty pocket of the active site, which do not only form hydrogen bonds with side chains or the protein backbone, but also with each other constituting a hydration network (Supplementary Fig. S8d).

The resolution in the active site is sufficient for complete atomic description of the coordinated $Ni^{2+}$ ions, including the carbamylated lysine. The protonation states of the active site residues are also represented in the map (Fig. 5b). One of the hydroxide molecules in the active site is essential as it performs the nucleophilic attack on urea while other molecules are displaced by urea and the closing of the mobile flap[3,4,8,9].

Comparison to the crystal structure of *K. aerogenes* of similar nominal resolution (1.9 Å) shows differences in the visualization of these features. This crystal structure was solved in absence of inhibitors or substrate such that the active site is also empty and the mobile flap in an open conformation (Fig. 5a). The details of the map provides finer details around the $Ni^{2+}$ ions in the cryo-EM map than the crystallographic data. The protonation of the histidines is clearly visible in the cryo-EM density (Fig. 5b). The positions of the side chains and the $Ni^{2+}$ ions in the active site are very similar to the *Y. enterocolitica* urease structure with a RMSD of 0.270 Å (Supplementary Table S5). The highest resolution *S. pasteurii* crystal structure was solved in presence of the inhibitor NBPT, which displaces the essential water molecules needed for the reaction from the active site. The closing of the mobile flap displaces the rest of the waters and brings the catalytic His323 closer to the active site. The tight packing of side chains prevents urea from entering the active site, efficiently blocking it (Fig. 5b, e). The active site residues and $Ni^{2+}$ ions have a RMSD of 0.293 Å between *S. pasteurii* and *Y. enterocolitica*.

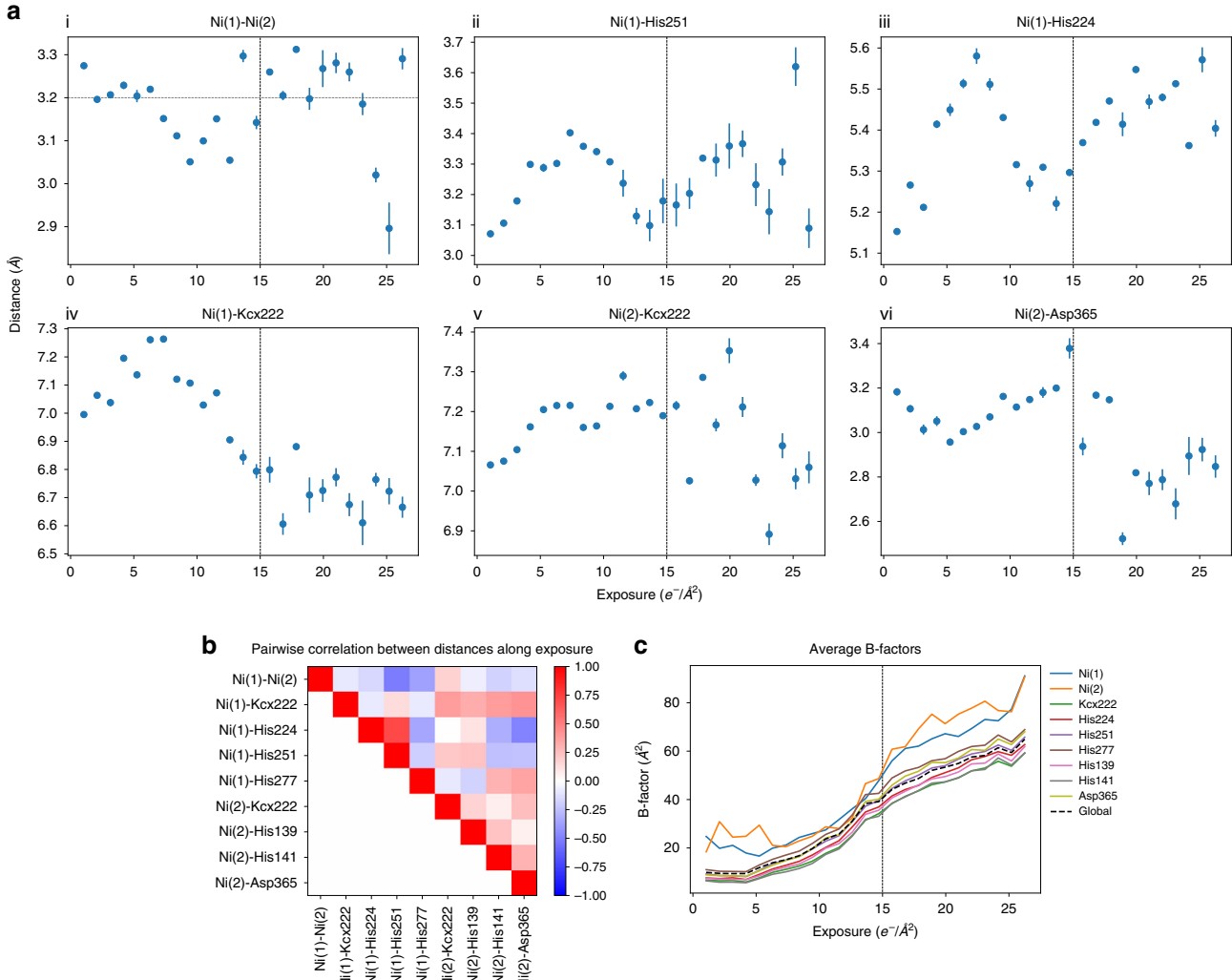

**Fig. 6 Radiation damage affects the distance between residues in the active site. a** Distances between the $Ni^{2+}$ ions and selected residues involved in their coordination are plotted against the accumulated exposure. For each reconstruction calculated along the exposure, the model was refined, and distances measured. Dots indicate the average and error bars show ±one standard deviation across five refinement runs with different random seeds. Horizontal dashed line in **a**–**i** shows the distance in the model obtained from the full reconstruction with all frames. Vertical dashed lines show approximately the exposure at which the density for charged residues completely vanishes (see Supplementary Movie 2). **b** Correlation coefficients between distance changes along the exposure for selected residues involved in ion coordination. Distance plots shown in **a** are indicated with a star. **c** Average B-factors of selected residues and of the global structure (from five refinement runs) plotted against the accumulated exposure.

**Nickel atoms come closer together**. The distance between the $Ni^{2+}$ ions is 3.7 Å in X-ray structures of *K. aerogenes* and *S. pasteurii*, but only 3.2 Å in the *Y. enterocolitica* cryo-EM model (Fig. 5b, d). Short distances of 3.1–3.3 Å were described for *S. pasteurii* and *K. aerogenes* at high resolutions for structures in presence of β-Mercaptoethanol (β-ME)[29]. Knowing that metallic cores are particularly sensitive to radiation[30], we tried to determine the extent to which radiation damage can explain the shorter distance between the $Ni^{2+}$ ions. For this purpose, we generated per-frame reconstructions for the first 25 frames of our data collection, refined the model on each of them (see "Methods"), and measured the distances between the residues involved in ion coordination, shown in Fig. 6. Bayesian particle polishing[17] was run again before calculating each per-frame reconstruction. At the beginning of the exposure, in which the frames contribute more to the full reconstruction due to dose weighting[17,31], there is a trend of the ions coming closer together (Fig. 6a−i). While we cannot determine exactly how this arises from radiation damage, it is likely a result of several interactions in the active site changing simultaneously along the exposure. For example, both Ni(1)

and Ni(2) tend to come closer to the carbamylated Lys222 (Fig. 6a-iv, v) as Asp365 vanishes (Fig. 6a-vi), which can be seen in the Supplementary Movie 2. Aspartic acid is known to have its side chain damaged very early on[32]. The dynamic interplay between residues along the exposure (Fig. 6b) is likely due to the different rates at which specific types of bonds and residues are damaged[33]: first negatively charged residues, then positively charged ones followed by aromatic side chains, as also observed in Supplementary Movie 2. A possible explanation of how these events may account for the shorter distance between the $Ni^{2+}$ ions is then that their bridging hydroxide molecule becomes deprotonated into the oxide form, either by radiation damage directly or by local pH changes arising from it. The oxide form is known to have a more favorable ferromagnetic interaction with the two nickel ions[25], although not found in ureases under native conditions[34]. Furthermore, the B-factors suggests that some residues in the active site, and in particular the $Ni^{2+}$ ions, are indeed damaged more strongly than the rest of the protein, right from the beginning of the irradiation as shown in Fig. 6c. We note however that, in the present analysis, radiation damage

cannot be completely disentangled from other effects such as residual sample movement, which is especially difficult to correct in the initial frames of the exposure. The later part of the exposure must also be interpreted with caution, as atomic coordinates become less reliable, which is verified by the overall increase in B-factors in Fig. 6c and the error bars in Fig. 6a.

## Discussion

Large urease assemblies have been historically difficult to study by X-ray crystallography[22]. We have determined the structure of a dodecameric urease assembly, a metalloenzyme from the pathogen *Y. enterocolitica* at an overall resolution of 1.98 Å using cryo-EM. The collection of datasets with and without beam-image shift demonstrates the advantages of using this feature of modern TEMs and invites further investigations on the behavior of optical aberrations and specimen drift.

Our results demonstrate the feasibility of cryo-EM as a technique for obtaining structures of clinically relevant enzymes with sufficient quality for de novo model building and drug design. The cryo-EM map has allowed a detailed description of the active site and the oligomeric assembly. More specifically, we could observe the putative oligomerization loop that enables the dodecameric assembly, which was hypothesized to be responsible for the enhanced survival of *Y. enterocolitica* in highly acidic environments[2]. This urease is the first outside the Helicobacteraceae family, and therefore without an αβ subunit organization, to have a dodecameric assembly reported. What evolutionary events have led to this intriguing combination of subunit organization and quaternary structure are unknown.

Furthermore, in comparison to the *K. aerogenes* structure, which is at approximately the same nominal resolution, the cryo-EM map offers an improved representation of protons and $Ni^{2+}$ ions. A possible explanation is that the error in the phases derived in the X-ray structure determination grows faster towards the limit of observed diffraction. Another aspect to be considered is that X-rays and electrons probe different properties of matter, respectively the electron density and the integrated Coulomb potential. Our results prompt a more detailed investigation of these effects and how they affect the representation of features at high resolution.

Finally, we noticed that radiation damage can partially explain the shorter distance observed between the nickel atoms in the active site, possibly by a stronger ferromagnetic interaction after deprotonation of the bridging hydroxide molecule[25,34]. Given that ions and charged residues are damaged very early on in the exposure[30,32], as we have also observed, this effect cannot be neglected in structures derived from cryo-EM reconstructions. Novel direct electron detectors with increased sensitivity and higher frame rates may allow the investigation of radiation damage effects in more detail.

## Methods

**Protein expression and purification**. The *Y. enterocolitica* urease was purified for cryo-electron microscopy according to the protocol from Rokita et al.[35]. The strain was precultured overnight at 37 °C for 18 h in a medium containing 37 g/l of brain/heart infusion (Oxoid, CM0225), 50 μg/ml streptomycin sulfate (Applichem, A1852.0100), 35 μg/ml nalidixic acid (Applichem, A1894.0025), 50 μg/ml meso-diaminopimelic acid (Sigma, D1377) and 100 μM nickel(II) chloride hexahydrate (Sigma, N6136). 6 × 600 ml of expression cultures were inoculated at OD of 1 at 28 °C for 23 h. Cells were harvested by centrifugation and the cell pellet resuspended in 0.15 M NaCl, 50 mM Tris pH 8.0. The cell lysate was applied directly to a Sephacryl S-300 HR 26/60 column equilibrated with 150 mM NaCl, 50 mM TrisHCl pH 8.0. The active fractions as identified by a phenol-hypochlorite assay[36] were buffer-exchanged to 50 mM Tris pH 7.0 within a centrifugal filter unit (Sartorius, Vivaspin MWCO 50 kDa) and applied to a Mono Q HR 5/5 column pre-equilibrated with 50 mM TrisHCl, pH 7.0. The protein was eluted in 50 mM Tris pH 7.0 by a gradient to 1 M NaCl, concentrated on a centrifugal filter unit (Sartorius, Vivaspin MWCO 50 kDa) and purified by SEC as before. The purity of

the urease sample of the two preparations was verified on a 4%/12% SDS-PAGE and by mass spectroscopy.

**Sample preparation**. Approximately 3 μl of the 0.39 mg/ml urease solution were applied to glow-discharged Quantifoil R 1.2/1.3 300 mesh holey carbon grids[37]. After 3-s blotting, the grids were flash-frozen in liquid ethane, using a FEI Vitrobot IV (Thermo Fisher Scientific) with the environmental chamber set at 90% humidity and 20 °C temperature.

**Data acquisition**. Cryo-EM data were collected on a FEI Titan Krios (Thermo Fisher Scientific) transmission electron microscope, operated at 300 kV and equipped with a Quantum-LS imaging energy filter (GIF, 20 eV zero loss energy window; Gatan Inc.) and a K2 Summit direct electron detector (Gatan Inc.) operated in dose fractionation mode. Data acquisition was controlled by the SerialEM[15] software, performed in counting mode, with a 42 e$^-$/Å$^2$ total exposure fractioned into 40 frames over 8 s. The physical pixel size was 0.639 Å at the sample level. The data were pre-processed via the FOCUS package[38], including drift-correction and dose-weighting using MotionCor2[39] (grouping every 5 frames and using 3 × 3 tiles) and CTF estimation using CTFFIND4[40] (using information between 30 Å and 5 Å from the movie stacks). With these settings, we collected two datasets: one using beam-image shift[14], with three shots per grid hole, comprising 2243 movies, and a second one taking a single shot per hole, with 2252 movies. Detailed data collection information are given in the Supplementary Table S1.

**Image processing**. The two datasets were initially processed separately as shown in the flowchart of Supplementary Fig. S2. We excluded all movies whose resolution of CTF fitting was worse than 4 Å according to CTFFIND4, leaving 2197 movies in the multi-shot dataset or 2115 in the single-shot dataset for further processing. Using the template-free LoG-picker algorithm[16] we picked an initial set of 157,699 particle coordinates on the multi-shot dataset. These particles were extracted and subjected to one round of 2D classification with the aim of removing bad or false-positive particles. Best results in 2D classification were observed when enabling the RELION option "Ignore CTFs until first peak?". Selecting only the classes displaying views of urease with high resolution features, a new subset containing 60,271 particles was obtained. Using this subset, a first 3D map was obtained by the ab initio stochastic gradient descent algorithm[16,41] with and without tetrahedral symmetry imposed. The symmetric map was consistent with previously determined structures of ureases[22,42]. The particles were then subjected to 3D refinement using the map from the ab initio procedure as starting reference, resulting in a map at 2.6 Å resolution. We then generated new templates for particle picking by low-pass filtering the unsharpened map from this first 3D refinement to 20 Å and calculating evenly oriented 2D projections from it. These templates were then used for picking with Gautomatch (K. Zhang, http://www.mrc-lmb.cam.ac.uk/kzhang/), detecting 107,399 particle coordinates on the multi-shot dataset or 87,204 particle coordinates on the single-shot dataset. Visual inspection of randomly selected micrographs indicated this set of coordinates was better than that previously found by the LoG-picker, in the sense that it contained fewer false positives and more true particles. The newly extracted particles were then subjected to 2D and 3D classification procedures to get rid of false positive, damaged or broken particles, which yielded cleaner subsets with 62,884 (multi-shot) or 51,173 (single-shot) particles. Using the current best map from the multi-shot dataset as a starting reference, we then performed masked 3D refinements on the two datasets separately, interleaved with rounds of CTF refinement and Bayesian particle polishing[17]. More specifically, we refined defocus per particle, astigmatism per micrograph and beam tilt globally in CTF refinement. In the multi-shot dataset, each of the three relative shooting targets were assigned a different class for separate beam tilt refinement. The parameters for Bayesian particle polishing were trained separately on ~5000 particles from each dataset at this stage. Each dataset yielded refined maps at 2.10 Å (multi-shot) and 2.20 Å (single-shot) resolution. Best results, however, were obtained when merging the particles picked by template-matching on each dataset (194,603 particles in total) and processing them altogether. After 2D classification, 141,069 particles remained (Supplementary Fig. S1b), and after 3D classification, there were 119,020 particles (Supplementary Fig. 1c). CTF refinement was then performed using four beam tilt classes, with the particles from the single-shot dataset belonging to a new, fourth class (Supplementary Table S2). Defocus and astigmatism were both refined per particle this time, resulting in a map at resolution of 2.05 Å. We compared polishing the full merged dataset at once and each beam tilt class separately, to verify if there were differences in the patterns of particle motion. For training the polishing parameters ~10,000 particles were used in each case this time. Although we did observe different statistics of particle motion (Supplementary Table S3), resolution and overall quality of the map did not improve further by performing either merged or separate polishing of the different shots. Finally, correction of third-order aberrations in RELION-3.1[19] (Supplementary Fig. S3) followed by local 3D refinement resulted in a global map resolution of 1.98 Å. The angular distribution efficiency was estimated using the program cryoEF using a box size of 512 voxels[20].

All resolution estimates reported were obtained by considering the 0.143 threshold[43] on the Fourier shell correlation (FSC) curve[44] between independently refined half-maps[45]. A solvent-excluding mask was generated by low-pass filtering

the maps to 12 Å, binarizing the filtered map and adding a soft edge consisting of a cosine-shaped falloff to zero. The FSC curve was corrected for artificial correlations introduced by the mask[46]. Local resolution was estimated using the approach implemented in RELION[47].

**Model building, refinement, and analysis**. After refinement of the map to high resolution it had to be flipped in UCSF Chimera[48] to match the correct handedness. The biological assembly from the crystal structure of *Y. enterocolitica* urease (Supplementary Note 1) was rigid-body fitted to the map in Chimera. The non-crystallographic symmetry (NCS) was calculated with PHENIX v1.17[49] from the crystal structure. Only using the hetero-trimer of the three proteins, backbone and side chains were built, corrected or confirmed in Coot[50]. After several rounds of manual refinement of the model in Coot, applying NCS and real-space refinement in PHENIX, the model comprised side chains of residues 1–100 of ureA, 31–162 of ureB, and 2–327, 335–572 of ureC. Residues 328–334 are disordered and could not be modeled with confidence. NCS constraints were not used during final refinements as to include alternative side chain conformations. Waters were built manually and refined in PHENIX and were added to the closest chain by the program phenix.sort_hetatoms. The quality of the refinement was assessed by the PHENIX cryo-EM Validation tool (Table 1)[21].

**Structure analysis**. The electrostatic potential was calculated using the APBS plugin in PyMOL[51]. The ConSurf server[23,24] was used to find 150 sequences per urease protein for alignment with ClustalW[52] and calculate conservation per residue. The "sample the list of homologs" option was used to get a diverse representation across all species. The sequence identity of the 150 sequences was determined using BLSM62 in Geneious[53]. The PDBePISA server (https://www.ebi.ac.uk/pdbe/pisa/) was used to find and calculate interface areas[54].

**Radiation damage analysis**. Bayesian particle polishing in RELION[16,17] was carried out on a sliding-window basis along the exposure, including 5 frames at a time, starting from frame 1 up to frame 25. Half-set reconstructions were then created from each polished particle stack and post-processed using the same mask as that applied to the reconstruction from all frames. On each post-processed reconstruction, real space refinement of chain C (α-subunit containing the active site) from the full reconstruction was carried out in PHENIX[55] for 5 macro-cycles. This procedure was repeated five times with different random seeds. Distance between residues in the resulting refined models were calculated using BioPython[56] and plotted using the NumPy (https://www.numpy.org) and Matplotlib (https://www.matplotlib.org) Python modules.

**Structural representations and figure generation**. Protein structural representations were generated using the following software: UCSF Chimera[48], ChimeraX[57], and PyMol (Schrödinger, LLC), with the aid of the Inkscape, Adobe Illustrator and Adobe Photoshop (Adobe Inc.) programs for creating figures.

**Reporting summary**. Further information on research design is available in the Nature Research Reporting Summary linked to this article.

## Data availability
The model has been deposited at the PDB under accession code 6YL3. The map has been deposited at the EMDB under accession code EMD-10835. Raw electron microscopy data is deposited in EMPIAR, accession code EMPIAR-10389. All other data supporting the findings of this study are available from the authors upon request.

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

## Acknowledgements

The authors would like to thank L. Kovacik and K. Goldie for assistance in data collection and S. Klumpe, A. Nunes-Alves, R. Ligabue-Braun, and T. Nakane for discussions. Cryo-EM data processing calculations were performed at sciCORE (http://scicore.unibas.ch/) scientific computing center at the University of Basel. R.D.R. and L.A. acknowledge funding from the Fellowships for Excellence program sponsored by the Werner-Siemens Foundation and the University of Basel. This work was in part supported by the Swiss National Science Foundation (grants 177195 and 185544, NCCR TransCure).

## Author contributions

R.D.R. and R.A. performed the cryo-EM experiments and data analysis. L.A. built and analyzed the atomic model. R.P.J. performed X-ray crystallography experiments. M.A.M. expressed and purified the protein. P.R. prepared and screened EM samples. J.Z. performed the higher-order aberration corrections and analysis. T.S., T.M., and H.S. initiated and supervised the project. R.D.R., L.A., R.A., T.M., and H.S. wrote the manuscript with assistance from all authors.

## Competing interests

The authors declare no competing interests.
