## [Peer Review File · Nature Communications]

REVIEWER COMMENTS

Reviewer #1 (Remarks to the Author):

The paper by Timm Maier and co-workers describes the obtainment of a high resolution ($< 2 \text{ \AA}$) structure of urease from *Yersinia enterocolitica* using cryo-EM. It is a remarkable result that should be published. The main conclusions of the work are that cryo-EM can be used to obtain valuable structural data for ureases from bacterial human pathogens in the absence of an otherwise critical crystallisation step, with the aim to support drug development for this virulence factor in several diseases caused by ureolytic microorganisms. This result is of extraordinary novelty and potential and will certainly be of interest to the wider scientific community working towards drug design and structure-activity relationships in critical enzymes, potentially changing the future approach to these goals. The proven capability to extract quasi-atomic molecular details using cryo-EM and not X-ray crystal diffraction opens new fields in the pursue of novel drug development. The paper should be published with minor revisions as I indicate below, to improve the manuscript impact:

Line 48

Add Mazzei, L.; Musiani, F.; Ciurli, S. Zamble, D., Rowinska-Zyrek, M., Kozłowski, H., Eds.; Royal Society of Chemistry: Cambridge, 2017; pp 60–97.

Line 52

Add Mazzei, L.; Musiani, F.; Ciurli, S. Zamble, D., Rowinska-Zyrek, M., Kozłowski, H., Eds.; Royal Society of Chemistry: Cambridge, 2017; pp 60–97.

Line 54

The reference Mazzei et al. 2017 should be substituted by Mazzei, L.; Musiani, F.; Ciurli, S. Zamble, D., Rowinska-Zyrek, M., Kozłowski, H., Eds.; Royal Society of Chemistry: Cambridge, 2017; pp 60–97.

Line 57

Add Mazzei, L.; Musiani, F.; Ciurli, S. Zamble, D., Rowinska-Zyrek, M., Kozłowski, H., Eds.; Royal Society of Chemistry: Cambridge, 2017; pp 60–97.

Lines 59-63

Add that the two Ni ions are bridged by a hydroxide ion that acts as the nucleophile, and add the following two references here to support this claim: Mazzei, L.; Cianci, M.; Benini, S.; Ciurli, S. Chemistry 2019, 25 (52), 12145–12158, and Mazzei, L.; Cianci, M.; Benini, S.; Ciurli, S. Angew. Chem. 2019, 131 (22), 7493–7497.

Lines 60-62

This is not correct: the current hypothesis of the mechanism entails that, after the nucleophilic attack on the carbonyl carbon of urea, a proton is transferred from the bridging hydroxide to the distal NH₂ group, not coordinated to Ni(II), yielding a nascent ammonia molecule.

Line 66

Add Mazzei, L.; Musiani, F.; Ciurli, S. Zamble, D., Rowinska-Zyrek, M., Kozłowski, H., Eds.; Royal Society of Chemistry: Cambridge, 2017; pp 60–97.

Line 75

Add Mazzei, L.; Musiani, F.; Ciurli, S. Zamble, D., Rowinska-Zyrek, M., Kozłowski, H., Eds.; Royal Society of Chemistry: Cambridge, 2017; pp 60–97.

Lines 200-204

This is common also in crystal structures and should be mentioned with reference to the pH dependence of this conformational movement, as explained in Mazzei, L.; Cianci, M.; Benini, S.; Ciurli,

S. Chemistry 2019, 25 (52), 12145–12158.

Lines 255-257

Change HIS, ASP, MET and LYS to His, Asp, Met and Lys here and everywhere

Line 262

Add Mazzei, L.; Musiani, F.; Ciurli, S. Zamble, D., Rowinska-Zyrek, M., Kozłowski, H., Eds.; Royal Society of Chemistry: Cambridge, 2017; pp 60–97.

Lines 262-263

See comment above (lines 60-62) for the description of the mechanism; change accordingly.

Lines 264-265

This is wrong: in the current most accepted mechanistic hypothesis, His325 acts as a base in its neutral form, which on one side allows the flap to close and on the other allows for the stabilization of the C(urea)-NH₃⁺ group to be stabilized after proton transfer from the bridging hydroxide after it has formed the O-C(urea) bond following nucleophilic attack.

Lines 265-266

Remove the reference to Kappaun and add Mazzei, L.; Musiani, F.; Ciurli, S. Zamble, D., Rowinska-Zyrek, M., Kozłowski, H., Eds.; Royal Society of Chemistry: Cambridge, 2017; pp 60–97.

Line 267

Add Mazzei, L.; Musiani, F.; Ciurli, S. Zamble, D., Rowinska-Zyrek, M., Kozłowski, H., Eds.; Royal Society of Chemistry: Cambridge, 2017; pp 60–97 and Mazzei, L.; Cianci, M.; Benini, S.; Ciurli, S. Chemistry 2019, 25 (52), 12145–12158, and Mazzei, L.; Cianci, M.; Benini, S.; Ciurli, S. Angew. Chem. 2019, 131 (22), 7493–7497.

Lines 269-271

pH is an important parameter to determine the flap conformation as explained in Mazzei, L.; Cianci, M.; Benini, S.; Ciurli, S. Chemistry 2019, 25 (52), 12145–12158. Discuss the pH of the sample studied.

Line 278

Remove this reference and add Mazzei, L.; Cianci, M.; Benini, S.; Ciurli, S. Chemistry 2019, 25 (52), 12145–12158; Mazzei, L.; Cianci, M.; Benini, S.; Ciurli, S. Angew. Chem. 2019, 131 (22), 7493–7497; Mazzei, L.; Musiani, F.; Ciurli, S. Zamble, D., Rowinska-Zyrek, M., Kozłowski, H., Eds.; Royal Society of Chemistry: Cambridge, 2017; pp 60–97; Maroney, M. J.; Ciurli, S. Chem Rev 2014, 114 (8), 4206–4228.

Lines 282-284

Explain how the protons are visualized; if, as I understand, the position of the protons is deduced from the H-bonds network derived from inter-residue interactions, the statement should be corrected. Only neutron scattering can detect protons.

Lines 286-291

The statement is not correct: there are other crystal structures of *S. pasteurii* urease in the native state, in the absence of substrate, and those should be mentioned, discussed and compared: PDB code 2UBP @ 2.00 Å (Benini, S.; Rypniewski, W. R.; Wilson, K. S.; Miletto, S.; Ciurli, S.; Mangani, S. Structure 1999, 7 (2), 205–216) and PDB code 4CEU @ 1.58 Å (Benini, S.; Cianci, M.; Mazzei, L.; Ciurli, S. J Biol Inorg Chem 2014, 19 (8), 1243–1261).

Lines 293-316

A possibility for the short distance between the Ni(II) ions that should be discussed is the

deprotonation of the bridging hydroxide to the oxide form, either by some sort of radiation damage or by local pH effects.

Line 616

Change *K. aerogenes* with *S. pasteurii* (PDB codes 2UBP and 3UBP are related to *S. pasteurii* and not *K. aerogenes*)

For all figures

The software used for production of the structural representation of protein pictures should be indicated.

Compliments!

Sincerely,
Stefano Ciurli

Reviewer #2 (Remarks to the Author):

In this manuscript, Righetto et al. describe the structure of *Yersinia enterocolitica* urease at high resolution, obtained by cryo-EM. Their major findings include the unusual dodecameric organization of the urease complex, the mapping of regions associated with this organization, and the analysis of radiation effects upon the description of metallocenters. So far the dodecameric organization has not been observed outside of the *Helicobacteriaceae* family, making the authors' findings highly attractive. The bacterial survival under acidic conditions seems to drive the recurrence of dodecameric ureases, as speculated by the authors. This high-resolution structure is not only relevant in the context of cryo-EM state of the art applications, but also due to the observations derived from the methods involved in its obtaining and analysis. The conclusions are solidly supported by the results presented, and the methods are described thoroughly, allowing for prompt reproduction of the experiments. This manuscript is well polished in its current form, providing intriguing insights into the evolution of an evolutionarily peculiar enzyme, urease.

Specific points:

- Figure 1, panel B, shows the isoform JBURE-II of *C. ensiformis* urease with 725 aa. It should be replaced by the full length urease, with 840 amino acid residues (doi:10.1016/j.bbapap.2011.07.022)
- Supplementary figure 5, as it appears in the manuscript, is blurred .
- Supplementary figure 6 is confusing. The same color (gray) is used to indicate sequences that did not align and protein stretches not resolved by the Cryo-EM. Please use different colors to illustrate these details.
- Please check the reference list – some references show errors in the authors' name, such as Kappaun et al, 2018.
- Yersinia enterocolitica* urease has been purified and biochemically characterized previously. At least one reference should be given.

Reviewer #3 (Remarks to the Author):

Review of NCOMMS-20-23905

High-resolution cryo-EM structure of urease from the pathogen *Yersinia enterocolitica*

The authors present the structure of the *Y. enterocolitica* urease by cryoEM. The structure is important, well done and the model is nicely supported by the experimental data. The authors have also provided extensive analysis of the details of the structure determination, and point to some interesting features that are probably related to radiation damage, all of which are very welcome. I recommend publication without delay after some minor revisions.

Specific points to address

It's not clear whether the changes presented in Fig. 6 are due to movement of the sample, global radiation damage, or specific damage to the active site. The authors should address this point in the text, but in a way that does not require additional experiments. Please also include a global B-factor vs frame plot, using the per-frame reconstructions already calculated, in the supplement to help clarify if the changes presented in fig 6 are site specific or part of the global degradation of the map with dose.

Other minor corrections

L33

Suggest you remove the "better than" and replace with something less provocative like "sub" or just 2 Å. I think the authors would be hard pressed to demonstrate the difference between a 1.98 vs a 2.00 vs a 2.05 Å map so they shouldn't place such value judgements in the abstract.

L39 "our data" avoid possessive tense here - it's the world's data. Perhaps "The structure"

L83 "extremely acidic" please provide pH or range

L86 unless the authors wish to demonstrate the significance of the 3 digit of precision, I suggest you round this to 2.0. You are welcome to include as many digits as you like when you deposit it in the database.

L138 not clear what is meant by "annealing". movement?

L92 predicate missing

L93 Last sentence belongs in the discussion or just omit. You have not actually discovered a drug in this work so should refrain from drug-discovery claims.

L167 to *a* local

Fig 5 The mesh is too dense to see through. Reduce line width?

L345 The last sentence is not clear and not supported by the paper. Radiation damage in this context is dose dependent, not time dependent. Rerword or omit.

L386 Include which grids were used for each structure, hole size etc. A citation to Ermantraut et al. 1998 should be added.

L587 "(left)" should be moved to before "filtered"

SupL13 typo '

SupL17 typo '

Sup Fig 2 the B-factor in the lower right is not physical and probably meaningless. Suggest omission.

Sup Fig 9 mesh is again too dense

Sup Table 3 " $\text{\AA}/\text{dose}$ " need units of dose here.

REVIEWER COMMENTS

Reviewer #1 (Remarks to the Author):

The paper by Timm Maier and co-workers describes the obtainment of a high resolution ($< 2 \text{ \AA}$) structure of urease from *Yersinia enterocolitica* using cryo-EM. It is a remarkable result that should be published. The main conclusions of the work are that cryo-EM can be used to obtain valuable structural data for ureases from bacterial human pathogens in the absence of an otherwise critical crystallisation step, with the aim to support drug development for this virulence factor in several diseases caused by ureolytic microorganisms. This result is of extraordinary novelty and potential and will certainly be of interest to the wider scientific community working towards drug design and structure-activity relationships in critical enzymes, potentially changing the future approach to these goals. The proven capability to extract quasi-atomic molecular details using cryo-EM and not X-ray crystal diffraction opens new fields in the pursue of novel drug development. The paper should be published with minor revisions as I indicate below, to improve the manuscript impact:

Line 48

Add Mazzei, L.; Musiani, F.; Ciurli, S. Zamble, D., Rowinska-Zyrek, M., Kozłowski, H., Eds.; Royal Society of Chemistry: Cambridge, 2017; pp 60–97.

The indicated reference has been added.

Line 52

Add Mazzei, L.; Musiani, F.; Ciurli, S. Zamble, D., Rowinska-Zyrek, M., Kozłowski, H., Eds.; Royal Society of Chemistry: Cambridge, 2017; pp 60–97.

The indicated reference has been added (now line 53).

Line 54

The reference Mazzei et al. 2017 should be substituted by Mazzei, L.; Musiani, F.; Ciurli, S. Zamble, D., Rowinska-Zyrek, M., Kozłowski, H., Eds.; Royal Society of Chemistry: Cambridge, 2017; pp 60–97.

The indicated reference has been replaced (now line 55).

Line 57

Add Mazzei, L.; Musiani, F.; Ciurli, S. Zamble, D., Rowinska-Zyrek, M., Kozłowski, H., Eds.; Royal Society of Chemistry: Cambridge, 2017; pp 60–97.

The indicated reference has been added (now line 59).

Lines 59-63

Add that the two Ni ions are bridged by a hydroxide ion that acts as the nucleophile, and add the following two references here to support this claim: Mazzei, L.; Cianci, M.; Benini, S.; Ciurli, S.

Chemistry 2019, 25 (52), 12145–12158, and Mazzei, L.; Cianci, M.; Benini, S.; Ciurli, S. Angew. Chem. 2019, 131 (22), 7493–7497.

We have included a sentence in the manuscript to clarify this point, with the indicated references (now lines 60-61).

Lines 60-62

This is not correct: the current hypothesis of the mechanism entails that, after the nucleophilic attack on the carbonyl carbon of urea, a proton is transferred from the bridging hydroxide to the distal NH₂ group, not coordinated to Ni(II), yielding a nascent ammonia molecule.

We have corrected our statement to reflect the current hypothesis on mechanism (now lines 63-64).

Line 66

Add Mazzei, L.; Musiani, F.; Ciurli, S. Zamble, D., Rowinska-Zyrek, M., Kozłowski, H., Eds.; Royal Society of Chemistry: Cambridge, 2017; pp 60–97.

The indicated reference has been added (now line 69).

Line 75

Add Mazzei, L.; Musiani, F.; Ciurli, S. Zamble, D., Rowinska-Zyrek, M., Kozłowski, H., Eds.; Royal Society of Chemistry: Cambridge, 2017; pp 60–97.

The indicated reference has been added (now line 78).

Lines 200-204

This is common also in crystal structures and should be mentioned with reference to the pH dependence of this conformational movement, as explained in Mazzei, L.; Cianci, M.; Benini, S.; Ciurli, S. Chemistry 2019, 25 (52), 12145–12158.

A clarification on the relationship between pH and the position of the mobile flap has been included, with the indicated reference (now lines 209-211).

Lines 255-257

Change HIS, ASP, MET and LYS to His, Asp, Met and Lys here and everywhere

All amino acid three-letter codes throughout the manuscript, including figures and the supplementary material have been formatted as requested (now lines 263-265, also Fig. 6 and further, all changes highlighted).

Line 262

Add Mazzei, L.; Musiani, F.; Ciurli, S. Zamble, D., Rowinska-Zyrek, M., Kozłowski, H., Eds.; Royal Society of Chemistry: Cambridge, 2017; pp 60–97.

The indicated reference has been added (now line 270).

Lines 262-263

See comment above (lines 60-62) for the description of the mechanism; change accordingly.

The mechanism explanation has been corrected as suggested (now lines 270-273).

Lines 264-265

This is wrong: in the current most accepted mechanistic hypothesis, His325 acts as a base in its neutral form, which on one side allows the flap to close and on the other allows for the stabilization of the C(urea)-NH₃⁺ group to be stabilized after proton transfer from the bridging hydroxide after it has formed the O-C(urea) bond following nucleophilic attack.

The explanation on the mobile flap position has been corrected (now lines 273-277).

Lines 265-266

Remove the reference to Kappaun and add Mazzei, L.; Musiani, F.; Ciurli, S. Zamble, D., Rowinska-Zyrek, M., Kozłowski, H., Eds.; Royal Society of Chemistry: Cambridge, 2017; pp 60–97.

The indicated reference has been replaced as suggested (now line 272-273). We have also included the reference to Mazzei, L.; Cianci, M.; Benini, S.; Ciurli, S. *Chemistry* 2019, 25 (52), 12145–12158 to support our explanation of the mobile flap mechanism.

Line 267

Add Mazzei, L.; Musiani, F.; Ciurli, S. Zamble, D., Rowinska-Zyrek, M., Kozłowski, H., Eds.; Royal Society of Chemistry: Cambridge, 2017; pp 60–97 and Mazzei, L.; Cianci, M.; Benini, S.; Ciurli, S. *Chemistry* 2019, 25 (52), 12145–12158, and Mazzei, L.; Cianci, M.; Benini, S.; Ciurli, S. *Angew. Chem.* 2019, 131 (22), 7493–7497.

The indicated references have been added (now line 275).

Lines 269-271

pH is an important parameter to determine the flap conformation as explained in Mazzei, L.; Cianci, M.; Benini, S.; Ciurli, S. *Chemistry* 2019, 25 (52), 12145–12158. Discuss the pH of the sample studied.

The position of the mobile flap has been discussed in light of the pH of our sample (7.0) (now lines 279-286).

Line 278

Remove this reference and add Mazzei, L.; Cianci, M.; Benini, S.; Ciurli, S. *Chemistry* 2019, 25 (52), 12145–12158; Mazzei, L.; Cianci, M.; Benini, S.; Ciurli, S. *Angew. Chem.* 2019, 131 (22), 7493–7497; Mazzei, L.; Musiani, F.; Ciurli, S. Zamble, D., Rowinska-Zyrek, M., Kozłowski, H., Eds.; Royal Society of Chemistry: Cambridge, 2017; pp 60–97; Maroney, M. J.; Ciurli, S. *Chem Rev* 2014, 114 (8), 4206–4228.

The indicated references have been added (now lines 294-295).

Lines 282-284

Explain how the protons are visualized; if, as I understand, the position of the protons is deduced from the H-bonds network derived from inter-residue interactions, the statement should be corrected. Only neutron scattering can detect protons.

As stated in lines 355-357 of our current manuscript (lines 342-344 in the initial submission), whereas X-rays are scattered by the electron density distribution of the sample, electrons are scattered by the atomic nuclei and by the electrostatic charge distribution, i.e. “the net charge of the electrons and nucleus of atoms” [1]. Transmission electron microscopy (TEM) images are therefore projections corresponding to the integrated Coulomb potential across the sample, rendering the reconstructed 3D maps from this technique fundamentally different from X-ray crystallography density maps. Protons are then visible by TEM as they do scatter electrons, but not X-rays. While our reconstruction is not of high enough resolution to visualize separated atoms as in true atomic resolution, at $\sim 2 \text{ \AA}$ “bumps” in the EM Coulomb potential map can already clearly indicate the position of hydrogen atoms, as is the case of the protonated H251 in Fig. 5b, for example. For further clarification, we refer the reviewer to recent atomic resolution protein structures solved by cryo-EM, which provide unambiguous visualization of protons as well as further details on their imaging [2,3]. We consider that our statements regarding the visualization of protonation states do not require correction.

[1] Marques, M. A., Purdy, M. D., & Yeager, M. (2019). CryoEM maps are full of potential. *Current Opinion in Structural Biology*, 58, 214–223. <https://doi.org/10.1016/j.sbi.2019.04.006>

[2] Yip, K. M., Fischer, N., Paknia, E., Chari, A., & Stark, H. (2020). Breaking the next Cryo-EM resolution barrier - Atomic resolution determination of proteins! *BioRxiv*, 2020.05.21.106740. <https://doi.org/10.1101/2020.05.21.106740>

[3] Nakane, T., Kotecha, A., Sente, A., McMullan, G., Masiulis, S., Brown, P. M. G. E., Grigoras, I. T., Malinauskaite, L., Malinauskas, T., Miebling, J., Yu, L., Karia, D., Pechnikova, E. V, de Jong, E., Keizer, J., Bischoff, M., McCormack, J., Tiemeijer, P., Hardwick, S. W., ... Scheres, S. (2020). Single-particle cryo-EM at atomic resolution. *BioRxiv*, 2020.05.22.110189. <https://doi.org/10.1101/2020.05.22.110189>

Lines 286-291

The statement is not correct: there are other crystal structures of *S. pasteurii* urease in the native state, in the absence of substrate, and those should be mentioned, discussed and compared: PDB code 2UBP @ 2.00 Å (Benini, S.; Rypniewski, W. R.; Wilson, K. S.; Miletto, S.; Ciurli, S.; Mangani, S. *Structure* 1999, 7 (2), 205–216) and PDB code 4CEU @ 1.58 Å (Benini, S.; Cianci, M.; Mazzei, L.; Ciurli, S. *J Biol Inorg Chem* 2014, 19 (8), 1243–1261).

We recognize that the wording in this section was misleading and tried to clarify at an earlier point in the manuscript (now lines 199-205 and 303). We are not aiming to provide a full list of all *S. pasteurii* and *K. aerogenes* urease structures, but choose the highest resolution urease structure available (PDB 5OL4) and an X-ray structure of similar nominal resolution (PDB 1EJW) to comparatively assess the quality of our reconstruction.

Lines 293-316

A possibility for the short distance between the Ni(II) ions that should be discussed is the deprotonation of the bridging hydroxide to the oxide form, either by some sort of radiation damage or by local pH effects.

This important possibility is now discussed in lines 334-339 and 371-373 of the revised manuscript.

Line 616

Change *K. aerogenes* with *S. pasteurii* (PDB codes 2UBP and 3UBP are related to *S. pasteurii* and not *K. aerogenes*)

The organism name has been corrected accordingly (now line 661).

For all figures

The software used for production of the structural representation of protein pictures should be indicated.

The software was indicated in the section "Structure analysis" of the Methods. We moved this to a new Methods section, "Structural representations and figure generation", for clarity (now lines 527-531).

Compliments!

Sincerely,
Stefano Ciurli

Reviewer #2 (Remarks to the Author):

In this manuscript, Righetto et al. describe the structure of *Yersinia enterocolitica* urease at high resolution, obtained by cryo-EM. Their major findings include the unusual dodecameric organization of the urease complex, the mapping of regions associated with this organization, and the analysis of radiation effects upon the description of metallocenters. So far the dodecameric organization has not been observed outside of the *Helicobacteriaceae* family, making the authors' findings highly attractive. The bacterial survival under acidic conditions seems to drive the recurrence of dodecameric ureases, as speculated by the authors. This high-resolution structure is not only relevant in the context of cryo-EM state of the art applications, but also due to the observations derived from the methods involved in its obtaining and analysis. The conclusions are solidly supported by the results presented, and the methods are described thoroughly, allowing for prompt reproduction of the experiments. This manuscript is well polished in its current form, providing intriguing insights into the evolution of an evolutionarily peculiar enzyme, urease.

Specific points:

-Figure 1, panel B, shows the isoform JBURE-II of *C. ensiformis* urease with 725 aa. It should be replaced by the full length urease, with 840 amino acid residues (doi:10.1016/j.bbapap.2011.07.022)

We have corrected Figure 1b to show full length JBU as indicated.

-Supplementary figure 5, as it appears in the manuscript, is blurred

The blurred Supp. Figure 5 has been replaced by a high-resolution version.

-Supplementary figure 6 is confusing. The same color (gray) is used to indicate sequences that did not align and protein stretches not resolved by the Cryo-EM. Please use different colors to illustrate these details.

We have changed the figure and the respective legend accordingly. Now, unmatched residues due to differences in sequence are shown in blue, while residues that have not been resolved in the cryo-EM structure are shown in gray.

-Please check the reference list – some references show errors in the authors' name, such as Kappaun et al, 2018.

Reference Kappaun et al, 2018 has been updated with the correct details (now line 579). The reference list has been edited and corrected.

-Yersinia enterocolitica urease has been purified and biochemically characterized previously. At least one reference should be given.

We extended a sentence in the introduction (now lines 88-89) to mention the previous biochemical characterization of *Y. enterocolitica* urease by Bhagat & Viridi, 2009.

Reviewer #3 (Remarks to the Author):

Review of NCOMMS-20-23905

High-resolution cryo-EM structure of urease from the pathogen Yersinia enterocolitica

The authors present the structure of the *Y. enterocolitica* urease by cryoEM. The structure is important, well done and the model is nicely supported by the experimental data. The authors have also provided extensive analysis of the details of the structure determination, and point to some interesting features that are probably related to radiation damage, all of which are very welcome. I recommend publication without delay after some minor revisions.

Specific points to address

It's not clear whether the changes presented in Fig. 6 are due to movement of the sample, global radiation damage, or specific damage to the active site. The authors should address this point in the text, but in a way that does not require additional experiments. Please also include a global B-factor vs frame plot, using the per-frame reconstructions already calculated, in the supplement to help clarify if the changes presented in fig 6 are site specific or part of the global degradation of the map with dose.

We rule out sample movement as a significant source of blurring in this analysis because each reconstruction has been calculated after a new particle polishing run (Zivanov et al, 2019) using

5-frame running averages along the exposure, as explained in the methods (lines 517-519 in the revised manuscript). This is now also clarified in lines 320-323. We have added the global B-factor vs. exposure plot as a black dashed line to Fig. 6c. This plot shows that the nickel ions are damaged more quickly than the rest of the protein structure right from the start of the irradiation as we have implied previously. This is now discussed in lines 339-342 and 373.

Other minor corrections

L33

Suggest you remove the "better than" and replace with something less provocative like "sub" or just 2 Å. I think the authors would be hard pressed to demonstrate the difference between a 1.98 vs a 2.00 vs a 2.05 Å map so they shouldn't place such value judgements in the abstract.

The abstract has been changed to simply mention a resolution of 2 Å.

L39 "our data" avoid possessive tense here - it's the world's data. Perhaps "The structure"

We have replaced "our data" with "The obtained structure" in the abstract.

L83 "extremely acidic" please provide pH or range

Y. enterocolitica urease has been shown to survive pH as low as 1.5. This is now mentioned explicitly in line 86 of the revised manuscript.

L86 unless the authors wish to demonstrate the significance of the 3 digit of precision, I suggest you round this to 2.0. You are welcome to include as many digits as you like when you deposit it in the database.

The sentence has been changed to simply mention an overall resolution of 2 Å (now line 90).

L138 not clear what is meant by "annealing". movement?

Yes, we meant the movement resulting from the annealing of the vitreous ice layer after being irradiated as described in Brilot et al, 2012. The sentence has been modified to clarify this point (now line 140).

L92 predicate missing

We have reworded the sentence to correct it (now lines 96-97).

L93 Last sentence belongs in the discussion or just omit. You have not actually discovered a drug in this work so should refrain from drug-discovery claims.

We have omitted the last sentence of the introduction. However, we kept the mention of the potential of cryo-EM for drug design in the discussion.

L167 to *a* local

The correction has been made (now line 169).

Fig 5 The mesh is too dense to see through. Reduce line width?

The line width was reduced to facilitate visualization.

L345 The last sentence is not clear and not supported by the paper. Radiation damage in this context is dose dependent, not time dependent. Reword or omit.

The sentence has been reworded for clarity, omitting the mention of temporal resolution (now lines 374-376).

L386 Include which grids were used for each structure, hole size etc. A citation to Ermantraut et al. 1998 should be added.

We have provided more information about the type of grid used and added the corresponding citation to Quantifoil grids (now line 416).

L587 "(left)" should be moved to before "filtered"

The figure legend has been changed as requested (now line 626).

SupL13 typo '

The typo has been corrected.

SupL17 typo '

The typo has been corrected.

Sup Fig 2 the B-factor in the lower right is not physical and probably meaningless. Suggest omission.

We changed the figure and the legend to clarify that this is the global sharpening (contrast restoration) B-factor estimated from the Guinier plot as described in Rosenthal & Henderson, 2003.

Sup Fig 9 mesh is again too dense

The line width was reduced to facilitate visualization.

Sup Table 3 "Å/dose" need units of dose here.

The units have been added accordingly (Å/e⁻).

REVIEWERS' COMMENTS:

Reviewer #1 (Remarks to the Author):

The authors have addressed all my comments. However, some sentences should be modified as follows, to be more precise, before final acceptance. If the authors comply with these requests, there will be no need to review the paper again before publication. Congratulations to the authors for a fine job!

Sincerely,
Stefano Ciurli

1.

Lines 63-65 of the revised manuscript: change "Urea first interacts with Ni(1) through its carbonyl oxygen, making urea more available for nucleophilic attacks. One of the amino nitrogens then binds to Ni(2) and a proton is transferred from the hydroxide ion to the distal amine in a nucleophilic attack on the carbonyl carbon of urea." into: "Urea first interacts with Ni(1) through its carbonyl oxygen, and following a conformational change of a mobile flap covering the active site, one of the amino nitrogens then binds to Ni(2), making the resulting metal-chelate urea molecule more available for nucleophilic attacks. Subsequently, the bridging hydroxide acts as the nucleophile attacking the urea C atom, and a proton is then transferred from the hydroxide ion to the distal NH₂ group, yielding a nascent ammonia molecule. The latter is released upon breaking of the C-NH₃⁺ bond, yielding carbamate as the byproduct, the latter spontaneously undergoing hydrolysis."

2.

Lines 212-214 of the revised manuscript: change "The position of the mobile flap has been related to the pH of the sample and both open and closed conformations have been observed in crystal structures (Mazzei et al., 2019a)." into "Both the open and the closed conformations of the mobile flap have been observed in crystal structures, stabilised at pH values lower and higher than the pK_a of the conserved His323, respectively (Mazzei et al., 2019a)."

3.

Lines 278-280 of the revised manuscript: change "After closing of the mobile flap, a proton is transferred from the bridging hydroxide ion molecule to the distal amine in a nucleophilic attack on the carbonyl carbon of urea." into "After closing of the mobile flap, the urea molecule chelates the two Ni ions in the active site, and following the nucleophilic attack by the bridging hydroxide onto the urea C atom, a proton is transferred to the distal amine group from the metal-bridging C-OH group, yielding a ammonia molecule after breakage of the resulting C-NH₃⁺ bond."

Reviewer #2 (Remarks to the Author):

All my comments were taken into consideration. The revised manuscript is much improved.

One point still needs to be addressed:

Please modify figure 1, panel B. As it looks now, it seems that JBUre-IIb (with 840 amino acid residues), is smaller than the bacterial ureases (with ca.800 residues, considering the sum of isolated subunits).

Celia R. Carlini

Reviewer #3 (Remarks to the Author):

High-resolution cryo-EM structure of urease from the pathogen *Yersinia enterocolitica*

The revisions are well done and do an excellent job of addressing all the points raised in review save one. The manuscript over all is certainly improved.

In the rebuttal letter response to Reviewer 3:

Reviewer 3:

It's not clear weather the changes presented in Fig. 6 are due to movement of the sample, global radiation damage, or specific damage to the active site. The authors should address this point in the text, but in a way that does not require additional experiments. Please also include a global B-factor vs frame plot, using the per-frame reconstructions already calculated, in the supplement to help clarify if the changes presented in fig 6 are site specific or part of the global degradation of the map with dose.

Response from Authors:

We rule out sample movement as a significant source of blurring in this analysis because each reconstruction has been calculated after a new particle polishing run (Zivanov et al, 2019) using 5-frame running averages along the exposure, as explained in the methods (lines 517-519 in the revised manuscript). This is now also clarified in lines 320-323. We have added the global B-factor vs. exposure plot as a black dashed line to Fig. 6c. This plot shows that the nickel ions are damaged more quickly than the rest of the protein structure right from the start of the irradiation as we have implied previously. This is now discussed in lines 339-342 and 373.

The statement

"We rule out sample movement as a significant source of blurring in this analysis because each reconstruction has been calculated after a new particle polishing run (Zivanov et al, 2019) using 5-frame running averages along the exposure, as explained in the methods (lines 517-519 in the revised manuscript)."

is demonstrably false, as can be seen from the authors' own data. Doing "particle polishing" runs does not prove that movement is not still causing blurring in the micrographs. In fact, the new Figure 6c shows just the opposite. The slope of the overall B-factor vs. exposure is slightly negative and then flat at the beginning of irradiation and non-linear throughout the exposure. If the blurring of the structure was due to radiation damage alone, the curve should be linear throughout the exposure (see Warkentin et al. 2014, for example). This indicates these data still suffer from significant blurring not caused by radiation damage even after particle polishing and motion correction. Even the B-factor for the Ni atoms gets better after $5 \text{ e}/\text{\AA}^2$, which cannot possibly be due to radiation damage. Thus the authors still need to revise and temper their claims about tracking the progress of radiation damage because it is not independent of other sources of exposure dependent degradation of image quality in the structure. This can be easily done by adding a statement to the text accordingly, and removing the claim on line 325 and anywhere else in the MS where necessary. Once this is complete it should be published without delay.

REVIEWERS' COMMENTS:

Reviewer #1 (Remarks to the Author):

The authors have addressed all my comments. However, some sentences should be modified as follows, to be more precise, before final acceptance. If the authors comply with these requests, there will be no need to review the paper again before publication. Congratulations to the authors for a fine job!

*Sincerely,
Stefano Ciurli*

1.

Lines 63-65 of the revised manuscript: change "Urea first interacts with Ni(1) through its carbonyl oxygen, making urea more available for nucleophilic attacks. One of the amino nitrogens then binds to Ni(2) and a proton is transferred from the hydroxide ion to the distal amine in a nucleophilic attack on the carbonyl carbon of urea." into: "Urea first interacts with Ni(1) through its carbonyl oxygen, and following a conformational change of a mobile flap covering the active site, one of the amino nitrogens then binds to Ni(2), making the resulting metal-chelate urea molecule more available for nucleophilic attacks. Subsequently, the bridging hydroxide acts as the nucleophile attacking the urea C atom, and a proton is then transferred from the hydroxide ion to the distal NH₂ group, yielding a nascent ammonia molecule. The latter is released upon breaking of the C-NH₃⁺ bond, yielding carbamate as the byproduct, the latter spontaneously undergoing hydrolysis."

The text has been changed according to the reviewer's suggestion.

2.

Lines 212-214 of the revised manuscript: change "The position of the mobile flap has been related to the pH of the sample and both open and closed conformations have been observed in crystal structures (Mazzei et al., 2019a)." into "Both the open and the closed conformations of the mobile flap have been observed in crystal structures, stabilised at pH values lower and higher than the pK_a of the conserved His323, respectively (Mazzei et al., 2019a)."

The text has been changed according to the reviewer's suggestion.

3.

Lines 278-280 of the revised manuscript: change "After closing of the mobile flap, a proton is transferred from the bridging hydroxide ion molecule to the distal amine in a nucleophilic attack on the carbonyl carbon of urea." into "After closing of the mobile flap, the urea molecule chelates the two Ni ions in the active site, and following the nucleophilic attack by the bridging hydroxide onto the urea C atom, a proton is transferred to the distal amine group from the metal-bridging C-OH group, yielding a ammonia molecule after breakage of the resulting C-NH₃⁺ bond."

The text has been changed according to the reviewer's suggestion.

Reviewer #2 (Remarks to the Author):

All my comments were taken into consideration. The revised manuscript is much improved.

One point still needs to be addressed:

Please modify figure 1, panel B. As it looks now, it seems that JBUre-IIb (with 840 amino acid residues), is smaller than the bacterial ureases (with ca.800 residues, considering the sum of isolated subunits).

Figure 1, panel B has been modified according to the reviewer's suggestion.

Celia R. Carlini

Reviewer #3 (Remarks to the Author):

Review 2 of NCOMMS-20-23905

High-resolution cryo-EM structure of urease from the pathogen *Yersinia enterocolitica*

The revisions are well done and do an excellent job of addressing all the points raised in review save one. The manuscript over all is certainly improved.

In the rebuttal letter response to Reviewer 3:

Reviewer 3:

It's not clear weather the changes presented in Fig. 6 are due to movement of the sample, global radiation damage, or specific damage to the active site. The authors should address this point in the text, but in a way that does not require additional experiments. Please also include a global B-factor vs frame plot, using the per-frame reconstructions already calculated, in the supplement to help clarify if the changes presented in fig 6 are site specific or part of the global degradation of the map with dose.

Response from Authors:

We rule out sample movement as a significant source of blurring in this analysis because each reconstruction has been calculated after a new particle polishing run (Zivanov et al, 2019) using 5-frame running averages along the exposure, as explained in the methods (lines 517-519 in the revised manuscript). This is now also clarified in lines 320-323. We have added the global B-factor vs. exposure plot as a black dashed line to Fig. 6c. This plot shows that the nickel ions are damaged more quickly than the rest of the protein structure right from the start of the irradiation as we have implied previously. This is now discussed in lines 339-342 and 373.

The statement

"We rule out sample movement as a significant source of blurring in this analysis because each reconstruction has been calculated after a new particle polishing run (Zivanov et al, 2019) using 5-frame running averages along the exposure, as explained in the methods (lines 517-519 in the revised manuscript)."

is demonstrably false, as can be seen from the authors' own data. Doing "particle polishing" runs does not prove that movement is not still causing blurring in the micrographs. In fact, the new Figure 6c shows just the opposite. The slope of the overall B-factor vs. exposure is slightly negative and then flat at the beginning of irradiation and non-linear throughout the exposure. If the blurring of the structure was due to radiation damage alone, the curve should be linear throughout the exposure (see Warkentin et al. 2014, for example). This indicates these data still suffer from significant blurring not caused by radiation damage even after particle polishing and motion correction. Even the B-factor for the Ni atoms gets better after 5 e/Å², which cannot possibly be due to radiation damage. Thus the authors still need to revise and temper their claims about tracking the progress of radiation damage because it is not independent of other sources of exposure dependent degradation of image quality in the structure. This can be easily done by adding a statement to the text accordingly, and removing the claim on line 325 and anywhere else in the MS where necessary. Once this is complete it should be published without delay.

We have removed the claim to sample movement not being a significant effect in our analysis. For further clarification, we have modified the end of the discussion about radiation damage to read as follows:

*"We note however that, in the present analysis, radiation damage cannot be completely disentangled from other effects such as residual sample movement, which is especially difficult to correct in the initial frames of the exposure. The later part of the exposure must also be interpreted with caution, as atomic coordinates become less reliable, which is verified by the overall increase in B-factors in **Fig. 6c** and the error bars in **Fig. 6a**."*